# Contrastive Representation Learning for Gaze Estimation

**Swati Jindal**                                                                          swjindal@ucsc.edu
**Roberto Manduchi**                                                              manduchi@soe.ucsc.edu
*University of California, Santa Cruz*
*Santa Cruz, CA, 95064, USA*

**Editor:** Editor's name

## Abstract

Self-supervised learning (SSL) has become prevalent for learning representations in computer vision. Notably, SSL exploits contrastive learning to encourage visual representations to be invariant under various image transformations. The task of gaze estimation, on the other hand, demands not just invariance to various appearances but also equivariance to the geometric transformations. In this work, we propose a simple contrastive representation learning framework for gaze estimation, named *Gaze Contrastive Learning (GazeCLR)*. *GazeCLR* exploits multi-view data to promote equivariance and relies on selected data augmentation techniques that do not alter gaze directions for invariance learning. Our experiments demonstrate the effectiveness of *GazeCLR* for several settings of the gaze estimation task. Particularly, our results show that *GazeCLR* improves the performance of cross-domain gaze estimation and yields as high as 17.2% relative improvement. Moreover, the *GazeCLR* framework is competitive with state-of-the-art representation learning methods for few-shot evaluation. The code and pre-trained models are available at https://github.com/jswati31/gazeclr.

**Keywords:** gaze estimation; representation learning; self-supervised learning

## 1. Introduction

Gaze represents the focus of human attention and serves as an essential cue for non-verbal communication. While specialized gaze trackers can accurately measure a user's gaze direction, there is substantial interest in gaze estimation using regular cameras. Although, learning gaze estimation models from images is challenging and needs to transcend multiple "nuisance" attributes such as facial features or head orientation to estimate gaze accurately.

In recent years, deep learning (Zhang et al., 2015, 2017; Krafka et al., 2016) has shown promising results for gaze estimation. In part, this success stems from the availability of large-scale annotated datasets. As a result, valuable datasets must contain a wide range of gaze directions, appearances, and head poses, which is laborious and time-consuming procedure. Also, gaze annotations are difficult to obtain, which makes the creation of large, representative datasets challenging (Ghosh et al., 2021). Therefore, methods that facilitate training with limited gaze annotations are highly desirable.

Self-supervised learning (SSL) has gained tremendous success over the past few years and emerged as a powerful tool for reducing over-reliance on human annotations (He et al., 2020; Chen et al., 2020a; Grill et al., 2020). Following a generally accepted paradigm, we consider a pre-training stage that requires no labels, followed by a fine-tuning stage using a relatively small number of labeled samples. SSL is an effective approach for pre-training,

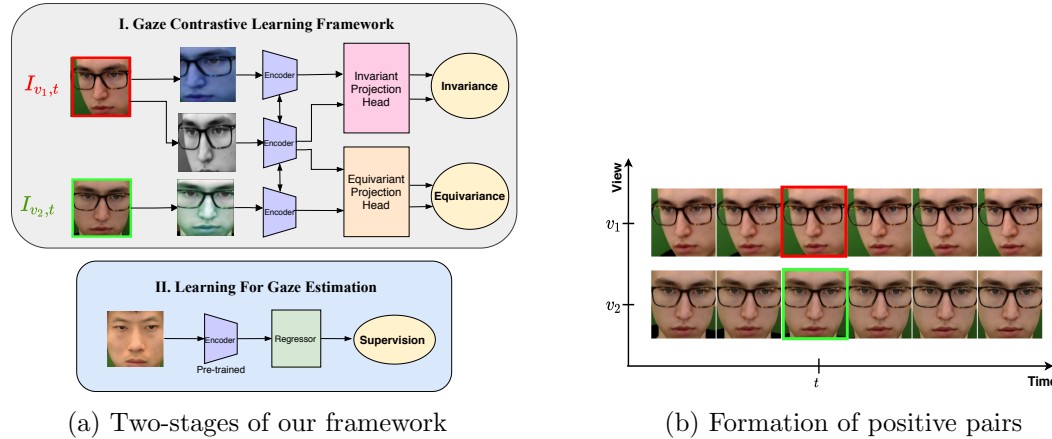

(a) Two-stages of our framework  (b) Formation of positive pairs

Figure 1: **Overall idea.** (a) The proposed two-stage learning framework for gaze estimation. Stage-I shows Gaze Contrastive Learning ($GazeCLR$) framework trained using only unlabeled data and learns both *invariance* and *equivariance* properties. In Stage-II, the pre-trained encoder is employed for gaze estimation task with small labeled data. (b) Two images (shown in **red** and **green**) captured at same time with different camera views are used to create both invariant and equivariant positive pairs.

where semantically meaningful representations are learned that can be seamlessly adapted during fine-tuning stage (Caron et al., 2018; Crawford and Pineau, 2019; Moriya et al., 2018). Specifically, a good pre-training would ensure that the embeddings for images associated with the same gaze direction are neighbors in the feature space, regardless of other non-relevant factors such as appearance. Arguably, this could accelerate the job of fine-tuning, possibly reducing the number of required labeled samples.

In this work, for SSL pre-training, we focus on *contrastive representation learning* (CRL), which aims to map "positive" pair samples to embeddings that are close to each other, while mapping "negative" pairs apart from each other (Chopra et al., 2005). A popular approach is to generate pairs by applying two different transformations (or augmentations) to an input image forming a positive pair, and different images forming negative pairs. This method encourages invariance in representations w.r.t. similar types of transformations, where these transformations are assumed to model "nuisance" effects.

However, obtaining the necessary and sufficient set of "positive" and "negative" pairs remains a non-trivial and unanswered challenge for a given task. This work attempts to answer this question for gaze estimation. Recent CRL-based methods encourage the representations to be invariant to *any* image transformation, many of which are not suitable for gaze estimation. For example, geometry-based image transformations (such as rotation) will change the gaze direction. In contrast, it is beneficial to have invariance to appearance, e.g., a person's identity, background, etc.

In this paper, we propose *Gaze Contrastive Learning* (or *GazeCLR*) framework – a simple CRL-based unsupervised pre-training approach for gaze estimation, i.e., a pre-training method requiring no gaze label data. In detail, our approach relies on *invariance* to image transforms (e.g., color jitter) that do not alter gaze direction and *equivariance* to camera viewpoint, which requires additional information of multi-view geometry, i.e.,

images of the same person should be obtained at the same time by two or more cameras from different locations.

For learning *equivariance*, we leverage the fact that in *a common reference system*, two or more synchronous images of the same person from different camera viewpoints are associated with the same gaze direction. The knowledge of the relative pose of each camera to the *common reference system* provides the relation of gaze directions defined in the respective camera space. In other words, gaze direction has an equivariant relationship to camera viewpoints. We claim that the requirement of using multiple cameras may be less onerous than obtaining gaze annotations for each image.

We use an existing multi-view gaze dataset EVE (Park et al., 2020) which provides video sequences captured from four calibrated and synchronized cameras and contains gaze annotations, which are obtained using a gaze tracking device (Tobii Pro AB, 2014). We neglect labels during pre-training and use them only for fine-tuning and evaluation. Observe that the relative camera pose information available with the EVE dataset is used *only* during the pre-training stage. Figure 1 presents an overview of the proposed idea.

To evaluate the *GazeCLR*, we perform self-supervised pre-training using the EVE dataset and transfer the learned representations for the gaze estimation task in various evaluation settings. We demonstrate the effectiveness of representations by showing that the proposed method achieves superior performance on both within-dataset and cross-dataset (such as MPIIGaze (Zhang et al., 2017) and Columbia (Smith et al., 2013)) evaluations by using only a small number of labeled samples for fine-tuning. Our major contributions are summarized as follows:

1. We propose a simple contrastive learning method for gaze estimation that relies on the *observation* that gaze direction is *invariant* under selected appearance transformations and *equivariant* to any two camera viewpoints.

2. We also argue to learn equivariant representations by taking advantage of the multi-view data that can be seamlessly collected using multiple cameras.

3. Our empirical evaluations show that *GazeCLR* yields improvements for various settings of gaze estimation and is competitive with existing supervised (Park et al., 2019) and unsupervised state-of-the-art gaze representation learning methods (Yu and Odobez, 2020; Sun et al., 2021).

## 2. Proposed Method

### 2.1. Stage-I: Gaze Contrastive Learning (*GazeCLR*) Framework

*GazeCLR* is a framework to train an *encoder* that learns embeddings to induce desired set of invariance and equivariance for the gaze estimation task. As stated earlier, the key intuition of *GazeCLR* is that we want to enforce invariance using selected appearance transformations (e.g., color jitter) and equivariance using synchronous images of the same person captured from multiple camera viewpoints. Similar to previous SSL approaches (Chen et al., 2020a; Spurr et al., 2021), we rely on the normalized temperature-scaled cross-entropy loss (NT-Xent)(Chen et al., 2020a) to encourage invariance or equivariance by maximizing the agreement between positive pairs and disagreement between the negative pairs. In

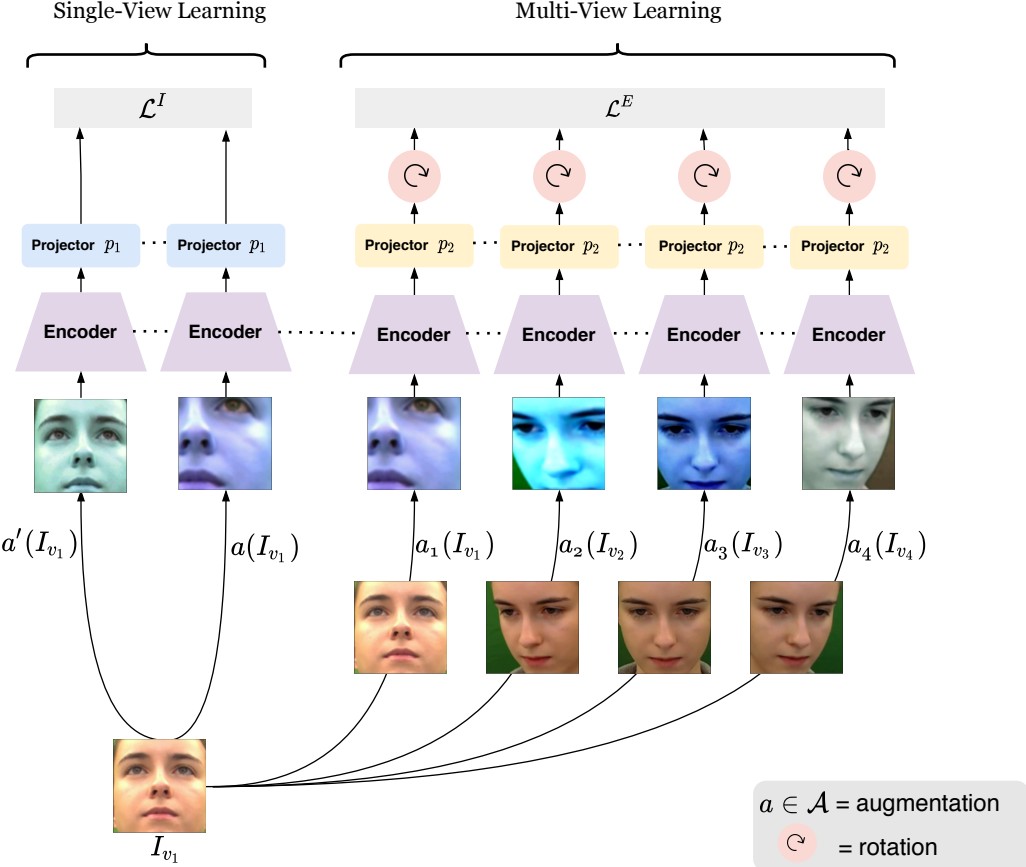

Figure 2: **Method schematic.** For synchronous view frames $\{I_{v_i}\}_{i=1}^4$, the above figure illustrates invariant and equivariant positive pairs anchored only for view $v_1$. The left branch shows *single-view* learning ($L^I$) and right branch illustrates *multi-view* learning using four views ($L^E$). All images (after augmentation, $a \in \mathcal{A}$) are passed through a shared CNN encoder network, followed by MLP projectors (either $p_1$ or $p_2$) depending on the type of input positive pair. The embeddings for multi-view learning are further multiplied by an appropriate rotation matrix. More details in Section 2.1.

particular, we devise two variants of NT-Xent loss, namely, $L^I$ for invariance and $L^E$ for equivariance.

The *GazeCLR* framework has three sub-modules: a CNN-based *encoder* and two *projection heads* based on MLP layers, as illustrated in Figure 2. The output of the encoder branches out into different *projection head* depending on the type of input positive pair. To abide by the invariance for gaze direction, we consider augmentations based on only *appearance* transformations denoted as $\mathcal{A}$.

Let $\{I_{v_i,t}\}_{i=1}^{|V|}$ be the synchronous frames for timestamp $t$ coming from different camera views (i.e., $\{v_i\}_{i=1}^{|V|}$), then we create the following positive pairs:

1. *Single-view positive pairs:* We apply two randomly sampled augmentations from $\mathcal{A}$ to create a single-view positive pair. Specifically, for any image $I_{v_i,t}$, at a given timestamp

$t$ and view $v_i$, we sample two augmentations $a$ and $a'$ from $\mathcal{A}$ and then $(a(I_{v_i,t}), a'(I_{v_i,t}))$ forms a positive pair to learn invariance. The left branch of Figure 2 shows one such positive pair for view $v_1$.

2. *Multi-view positive pairs:* We consider all unique pairs of camera viewpoints from the same timestamp $t$ and apply random augmentations from $\mathcal{A}$, i.e., $\{(a_i(I_{v_i,t}), a_j(I_{v_j,t})) \mid i, j \in \{1, \dots, |V|\} \mid i \neq j\}$. The corresponding outputs from the encoder are passed through projection head $p_2$ and multiplied by an appropriate rotation matrix to learn equivariance.

Next, to construct negative pairs, we do not sample them explicitly but use all other samples in the mini-batch as negative examples, similar to Chen et al. (2020a). The exact formulation of both loss functions $L^I$ and $L^E$ is described below. For brevity, we omit $t$ from $I_{v_i,t}$ and augmentation $a$ in the following subsections.

### 2.1.1. Single-View Learning

Recall, the goal of *single-view* learning is to induce invariance amongst representations under various appearance transformations. Let $v_i \in V$ be any view and $b \in [1, \dots, B]$ be the batch index. Given a batch size of $B$, we apply two augmentations to each sample in the batch yielding $2B$ augmented images, and for each sample, we have one positive pair and $(2B - 1)$ negative pairs stemming from remaining samples in the batch. Our encoder $E$ extracts representations for all $2B$ augmented images, which are further mapped by projection head $p_1(\cdot)$ yielding embeddings $(\{z_{v_i}^b, z_{v_i}'^b\}_{b=1}^B)$. With above notations, for any view $v_i$, the proposed invariance loss function $L^I$ associated with a positive pair $(z_{v_i}^b, z_{v_i}'^b)$ can be given as follows:

$$L^I(z_{v_i}^b, z_{v_i}'^b) = -\log \frac{\texttt{sim}(z_{v_i}^b, z_{v_i}'^b)}{\sum_{l=1}^{B} 1_{l \neq b} \texttt{sim}(z_{v_i}^b, z_{v_i}^l) + \sum_{l=1}^{B} \texttt{sim}(z_{v_i}^b, z_{v_i}'^l)} \tag{1}$$

where, $z_{v_i}^b = p_1(E(I_{v_i}^b))$, $z_{v_i}'^b = p_1(E(I_{v_i}'^b))$, $\texttt{sim}(r, s) = \exp\left(\dfrac{1}{\tau} \dfrac{r^T s}{||r|| \cdot ||s||}\right)$, $1_{[l \neq b]}$ is an indicator function and $\tau$ is the temperature coefficient parameter. It is worth noting that to minimize the loss in Eq. 1, it must hold that $z_{v_i}^b$ and $z_{v_i}'^b$ needs to be closer, which aligns with our goal of learning invariance to appearance transformations. One challenge, however, is the risk of collapse (e.g., the network could simply learn each person's identity). To avoid this, we create mini-batches such that all samples in a batch are taken from a single participant.

### 2.1.2. Multi-View Learning

We encourage equivariance in the gaze representations to different camera viewpoints through multi-view learning. To do so, we transform embeddings to a common reference system, chosen as the *screen reference system* used during the EVE data collection. Let $\{R_{C_{v_i}}^S\}$ be the rotation matrix relating the camera viewpoint $v_i$ with the screen reference system.

For each sample $I_{v_i}^b$ in a batch of size B, the positive pair is given as $(I_{v_i}^b, I_{v_j}^b)$ for two distinct camera viewpoints $(v_i, v_j)_{i \neq j}$. All images for viewpoints $v_i$ and $v_j$ are first augmented then passed through encoder $E$ and the projector head $p_2(\cdot)$ which gives embeddings

$\hat{z}_{v_i}^b, \hat{z}_{v_j}^b \in R^{3 \times d'}$. These embeddings are further multiplied by corresponding rotation matrices $R_{C_{v_i}}^S$ to project embeddings in the common (screen) reference system. We denote embeddings after rotation as $\{\bar{z}_{v_i}^b, \bar{z}_{v_j}^b\}_{b=1}^B$ such that $\bar{z}_{v_i}^b = R_{C_{v_i}}^S \hat{z}_{v_i}^b$. Therefore, for a batch of size B, our equivariant loss $L^E$ associated with the positive pair $(\bar{z}_{v_i}^b, \bar{z}_{v_j}^b)$ is as follows:

$$L^E(\bar{z}_{v_i}^b, \bar{z}_{v_j}^b) = -\log \frac{\mathtt{sim}(\bar{z}_{v_i}^b, \bar{z}_{v_j}^b)}{\sum_{l=1}^B 1_{[l \neq b]} \mathtt{sim}(\bar{z}_{v_i}^b, \bar{z}_{v_i}^l) + \sum_{l=1}^B \mathtt{sim}(\bar{z}_{v_i}^b, \bar{z}_{v_j}^l)} \tag{2}$$

**Overall loss function.** Given $|V|$ camera viewpoints, we apply both $L^I$ and $L^E$ loss functions to each view. Thus, our overall objective function for a batch size of B becomes:

$$L^O = \frac{1}{2B} \sum_{i=1}^{|V|} \sum_{b=1}^B \left( L^I(z_{v_i}^b, z_{v_i}'^b) + L^I(z_{v_i}'^b, z_{v_i}^b) + \sum_{j=1, j \neq i}^{|V|} L^E(\bar{z}_{v_i}^b, \bar{z}_{v_j}^b) \right) \tag{3}$$

### 2.2. Stage-II: Learning For Gaze Estimation

After pre-training, the encoder learned by the $GazeCLR$ framework is used for the task of gaze estimation and fine-tuned on a small labeled dataset. To this end, we remove both projection heads $p_1$ and $p_2$, and replace them with MLP regressor layers to predict 3D gaze direction. For training MLP regressor, we use the supervised loss function given as

$$L^{ang} = \frac{180}{\pi} \arccos \left( \frac{\boldsymbol{g} \cdot \hat{\boldsymbol{g}}}{||\boldsymbol{g}|| \cdot ||\hat{\boldsymbol{g}}||} \right) \tag{4}$$

where $\boldsymbol{g}$ and $\hat{\boldsymbol{g}}$ are the ground-truth and predicted gaze directions, respectively.

## 3. Experiments

We start by detailing the experimental setup (Sec. 3.1) followed by a brief explanation of considered baselines (Sec. 3.2). Next, we evaluate the performance of our pre-trained encoder and show that representations from $GazeCLR$ can help train an accurate gaze estimation model even with a relatively lesser amount of annotations. For this task, we consider the within-dataset setting (Sec. 3.3). We assess the transferable capability of our representations by evaluating them on different domains in linear layer training (frozen encoder) setting, where we considered only a few calibration samples from the test subject, as detailed in Sec. 3.4. Thereafter, we compare $GazeCLR$ with existing supervised (Park et al., 2019) and unsupervised (Yu and Odobez, 2020; Sun et al., 2021) pre-training methods in Sec. 3.5. Lastly, we probe the semantics of learned $GazeCLR$ representations using a well-known t-SNE visualization technique (in Sec. 3.6). Additional results and ablation studies are provided in appendix B and C.

### 3.1. Setup

We train our $GazeCLR$ framework on the EVE (Park et al., 2020) dataset, which has videos collected in a constrained indoor setting with four different synchronized and calibrated camera views. It has approximately 12 million frames collected from 54 participants with

natural eye movements. Following the splits considered by Park et al. (2020), there are 40 subjects in training and 6 subjects in the validation set. We discard the data of test subjects due to the non-availability of labels. We use training subjects for the pre-training stage, *without* using any gaze annotations. For the gaze estimation stage, we evaluate on the data of validation subjects to report the performance. We use all four camera views (i.e., $|V| = 4$) as well as the information about the relative pose between camera and screen ($R_C^S$) provided with the EVE dataset. Note that our framework can be extended to more number of camera views ($|V| > 4$) using ETH-XGaze (Zhang et al., 2020) dataset. In this paper, we consider pre-training only on EVE dataset as more views add on increased computational demand.

**Data pre-processing.** We use face images available in the EVE dataset, obtained after applying a data-normalization procedure (Sugano et al., 2014; Zhang et al., 2018). The normalization pipeline transforms the gaze annotation to a normalized camera space through a rotation matrix $M$. Note that we post-multiply $R_C^S$ with $M^{-1}$ as $R_C^S$ is defined w.r.t. the original camera reference frame, i.e., $\bar{z}_v = R_{C_v}^S (M)^{-1} \hat{z}_v$.

**Training details.** *GazeCLR* is trained using SGD optimizer with initial learning rate $= 0.03$, momentum $= 0.9$, and cosine annealing (Loshchilov and Hutter, 2016) for the learning rate decay. We use a single 1080 GeForce GTX GPU for training, with a batch size of 128, and train for 50K iterations. Our mini-batch is made up of samples from a single participant. The temperature coefficient $\tau$ is set to 0.1. For the augmentation transformations $\mathcal{A}$, we apply random spatial cropping and resizing, gaussian blur, color perturbation ($p = 0.8$) on brightness, contrast, saturation and hue, grayscale conversion ($p = 0.2$), and auto-contrast ($p = 0.5$).

All experiments use ResNet-18 (He et al., 2016) as the encoder network and take the output from the average pooling layer. The encoder is trained from scratch. Following Chen et al. (2020a), both projection heads $p_1(\cdot)$ and $p_2(\cdot)$ are two-layer MLP networks with ReLU non-linearity. The output dimensions for the first and second layers are 512 and 180, respectively. The input image size is $128 \times 128$.

We train the *GazeCLR* framework in two different settings: (i) *GazeCLR (Equiv)*: where we only consider equivariance through the loss function $L^E$ and (ii) *GazeCLR (Inv+Equiv)*: where we consider both invariance and equivariance with equal weights using the overall objective $L^O$. We present the performance of both training setups in all the considered experimental settings. Observe that, *GazeCLR (Inv)* trained with only $L^I$ loss function is equivalent to SimCLR (Chen et al., 2020a) baseline method.

### 3.2. Baselines

We compare our approach with six following baselines: (i) *w/o Pre-training*, i.e., an encoder is initialized using random weights, (ii) the vanilla *Autoencoder*, which has an encoder network that consists of the same encoder layers as *GazeCLR* and five DenseNet (Huang et al., 2017a) deconvolution blocks as decoder, and is trained with L2 loss, (iii) *Novel View Synthesis* (Rhodin et al., 2018) framework is trained on our dataset using the same architecture as the auto-encoder, (iv) BYOL (Grill et al., 2020), (v) SimCLR (Chen et al., 2020a) and (vi) *Fully-Supervised* is a ResNet-18 model trained on the whole EVE training data and represents possibly an upper bound for the performance of *GazeCLR*. For SimCLR

Table 1: **Within-dataset Evaluation.** We report the mean angular errors (MAE) in degrees for within-dataset evaluation for the gaze estimation task. The "EVE" shows the whole EVE data while "MiniEVE" indicates a small subset data. The Frozen column is ✓ if pre-trained encoder is frozen, otherwise fine-tuned ✗. The best performing method is shown in **bold** and second best is underlined.

| Method | Pre-Train Data | Task Data | Frozen | MAE ↓ (degrees) |
|---|---|---|---|---|
| w/o Pre-training | EVE | MiniEVE | ✗ | 8.47 |
| Autoencoder | EVE | MiniEVE | ✗ | 6.91 |
| Novel View Synthesis (Rhodin et al., 2018) | EVE | MiniEVE | ✓ | 6.79 |
| BYOL (Grill et al., 2020) | EVE | MiniEVE | ✗ | 8.35 |
| SIMCLR (Chen et al., 2020a) | EVE | MiniEVE | ✓ | 6.57 |
| **GazeCLR (Equiv)** | EVE | MiniEVE | ✓ | **4.83** |
| **GazeCLR (Inv+Equiv)** | EVE | MiniEVE | ✓ | 4.92 |
| Fully-Supervised | - | EVE | ✗ | 4.15 |

and BYOL, we use the same augmentation set as in our proposed method. For more experimental details, see appendix E.

### 3.3. Within-dataset Evaluation

For within-dataset evaluation, we perform pre-training on the training split of the EVE dataset without using labels. Then we adapt the pre-trained encoder for the gaze estimation on a small subset of labeled data. Precisely, we took five training subjects out of 40 (which form around 10% samples out of the whole EVE dataset) for the supervised gaze estimation stage and called it "MiniEVE". We validate on fixed subject data chosen from training subjects and report the final performance for validation subjects.

Table 1 shows the mean angular errors (in degrees) obtained for different pre-training baselines and the proposed *GazeCLR* method. To this end, we freeze the pre-trained encoder and simply train an MLP regressor using the "MiniEVE" dataset. Note that, for two baselines, Autoencoder and BYOL, we fine-tune the whole end-to-end framework along with the encoder as otherwise, they fail to converge when only their representations are used. We indicate this behavior in Table 1, using the *Frozen* column as ✓ if encoder is frozen otherwise as ✗.

We observe that our method *GazeCLR* outperforms other pre-training baseline methods by only training an MLP regressor on the small amount of labeled data ("MiniEVE" is ∼ 10% of whole data). Specifically, it can be seen that the performance achieved from *GazeCLR* helps in closing the gap with the fully-supervised baseline. Our method *GazeCLR (Inv+Equiv)* shows a relative improvement of 25.1% compared to the popular contrastive learning method SimCLR. Additionally, *GazeCLR (Equiv)* shows a boost of 26.4% relative improvement over the SimCLR approach, suggesting that equivariant representations are very effective for the gaze estimation task. We hypothesize that since we utilize similar augmentation strategies for creating both single-view and multi-view positive pairs, *GazeCLR (Equiv)* performs almost comparable to *GazeCLR (Inv+Equiv)*.

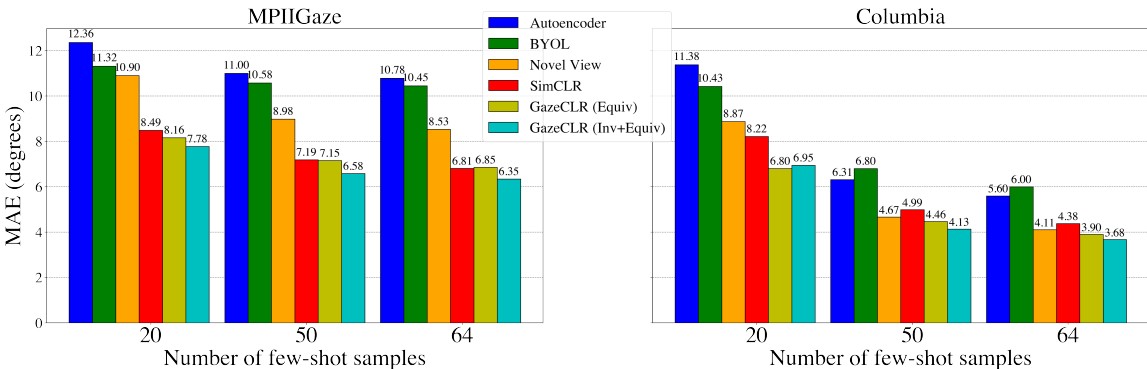

Figure 3: **Transfer Learning Evaluation.** Performance evaluation using *Linear Layer Training (LLT)* protocol for both MPIIGaze and Columbia dataset under different few-shot settings. Each bar is computed by averaging over 10 runs. Best viewed in color.

### 3.4. Transfer Learning/Cross-dataset Evaluation

We perform a cross-dataset evaluation using a few-shot personalized gaze estimation to further demonstrate the cross-data generalization capabilities of the learned representations. We evaluate *GazeCLR* representations on two domain datasets different from pre-training data: MPIIGaze (Zhang et al., 2015) and Columbia (Smith et al., 2013). **MPIIGaze** is a challenging dataset that has higher inter-subject variations. We use the standard evaluation subset MPIIFaceGaze (Zhang et al., 2017), containing around 37667 images captured from 15 subjects. The **Columbia** dataset consists of 5880 images collected from 56 subjects and is known to have high head pose variations.

To measure the quality of learned representations, we use *Linear Layer Training (LLT)* protocol, in which we freeze the trained encoder and learn a linear regressor on the target dataset. For this experiment, we investigate under a few-shot setting where we sample a few calibration samples from the test subject for adaptation and evaluate on the remaining samples of the same test subject.

Figure 3 shows the mean angular errors for LLT protocol on 20-shot, 50-shot, and 64-shot gaze estimation. We first extract the gaze representations of a few calibration samples for each subject and learn a linear model on top of these representations. We evaluate the trained model on the remaining samples of the subject. We repeat above 10 times for each subject on both datasets and report mean angular errors for the same in Figure 3.

Observe that both proposed *GazeCLR* variants outperform all other baselines in all few-shot settings for both datasets. Moreover, *GazeCLR(Equiv)* gives a relative improvement of around 17.2% over SimCLR with only 20 calibration samples for Columbia. We hypothesize that this behavior is due to high head-pose variations within Columbia, and it suggests that: a) learning equivariance over multi-views is beneficial for the GazeCLR framework in improving performance, and b) *GazeCLR* representations are relatively more generalizable for cross-domain datasets than other baselines.

Table 2: ***GazeCLR* vs Yu and Odobez (2020); Sun et al. (2021):** Comparison of *GazeCLR* with other unsupervised gaze representation learning methods (Yu and Odobez, 2020; Sun et al., 2021) for 50-shot gaze estimation. † denotes the method that uses additional head pose information. The metric reported is mean angular errors averaged over 10 runs (in degrees).

| Method | Pre-Train Data | MPIIGaze | Columbia |
|---|---|---|---|
| Yu and Odobez (2020)† | Columbia | - | 8.9 |
| Sun et al. (2021) | MPIIGaze | 8.5 | - |
| Sun et al. (2021) | Columbia | - | 7.0 |
| ***GazeCLR* (Equiv)** | EVE | 7.0 | **6.1** |
| ***GazeCLR* (Inv+Equiv)** | EVE | **6.5** | 6.6 |

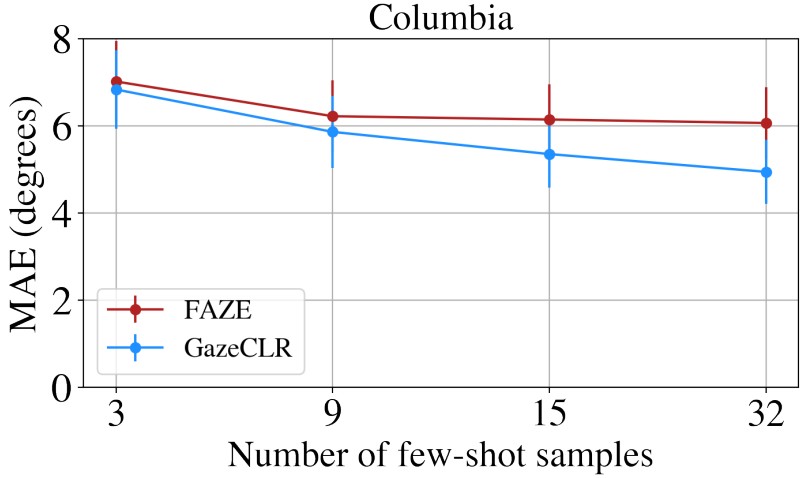

Figure 4: ***GazeCLR* vs FAZE (Park et al., 2019).** Comparison of *GazeCLR* with supervised pre-training baseline (FAZE) for various few-shot settings, on the Columbia dataset. The plot shows mean angular error (MAE, in degrees) and standard error bars *versus* number of few-shot samples, reported after 10 runs.

### 3.5. Comparison with state-of-the-art gaze representation learning

We further compare *GazeCLR* with existing state-of-the-art unsupervised (Yu and Odobez, 2020; Sun et al., 2021) and supervised (Park et al., 2019) gaze representation learning methods. For a fair comparison, we adopt the same evaluation protocols as used by these baseline methods and compare the *GazeCLR* performance against their performance.

***GazeCLR* vs. Unsupervised Pre-training (Yu and Odobez, 2020; Sun et al., 2021).** We follow the same evaluation protocol as (Yu and Odobez, 2020). 5-fold and leave-one-out (15-fold) evaluations are used for the Columbia and MPIIGaze datasets, respectively. In each fold, we freeze the *GazeCLR* encoder and extract representations for randomly selected 50 samples with annotations and learn a simple MLP-based gaze estimator on top of that. We repeat the performance evaluation 10 times and report mean angular errors in Table 2.

Note that previous methods (Yu and Odobez, 2020; Sun et al., 2021) exploit left and right eye patches to get SSL signal, whereas our approach relies on face patches obtained from multiple camera viewpoints.

In Table 2, we compare against the best-performing models of Yu and Odobez (2020) and Sun et al. (2021), for the 50-shot gaze estimation. Notice that our method outperforms baselines with absolute improvements of 2° and 0.9° on MPIIGaze and Columbia, respectively. It is worth emphasizing that our method is pre-trained on a different dataset than both evaluation datasets, unlike baseline approaches. Again, it illustrates the strength of our approach in learning semantically meaningful representations for generalizable to other domains. Moreover, note that Yu and Odobez (2020) use additional head-pose information, unlike our method.

***GazeCLR* vs. Supervised Pre-training (Park et al., 2019).** We evaluate the effectiveness of *GazeCLR* representations using the MAML framework (Finn et al., 2017), similar to FAZE (Park et al., 2019). For both *GazeCLR* and FAZE, we train a MAML-based gaze estimator on the representations for subjects from the GazeCapture (Krafka et al., 2016) dataset. Then, we adapt the gaze estimator model to each test subject of Columbia with $k$ calibration samples and test on the remaining samples. Figure 4 depicts the performance comparison of *GazeCLR* with FAZE (Park et al., 2019) for four different values of $k$. It can be seen that our method consistently outperforms supervised pre-training baseline FAZE, for all values of $k$. Notably, our framework uses *zero* labeled information to obtain gaze representations, unlike FAZE, which is pre-trained using $\sim 2M$ labeled samples from the GazeCapture dataset.

### 3.6. Visualization of Gaze Representations

To further investigate the quality of learned representations, we project the gaze representations into 2-dimensions using t-SNE (Van der Maaten and Hinton, 2008) algorithm as shown in Figure 5. In Fig 5(a), we compute 2D visualization of equivariant representations obtained after applying rotation matrices, i.e., $\bar{z}$. Projections in Fig 5(a) clearly demonstrate that gaze direction is invariant to the viewpoint, as images at the same timestamp from different views are mapped closer (shown with the same color border). In Fig 5(b), we apply t-SNE algorithm on gaze representations obtained at the output of encoder network, i.e., $z = \mathrm{E}(\cdot)$, for images from single camera viewpoint. Projections corresponding to roughly similar gaze directions are naturally clustered and highlighted with different background colors. Also, we observe clear patterns in the learned feature space where images within close vicinity are invariant to the subject's identity, showing invariance towards appearances.

## 4. Related Work

**Gaze Estimation.** Gaze estimation methods are built using large-scale datasets either having 2D target labels (Krafka et al., 2016; Huang et al., 2017b) or 3D gaze directions (Fischer et al., 2018b; Funes Mora et al., 2014; Zhang et al., 2015). Broadly, gaze estimation methods can be divided into two categories: appearance methods (Tan et al., 2002), which directly map image pixels to 3D gaze direction, and model methods which rely on eye-geometry (Hansen

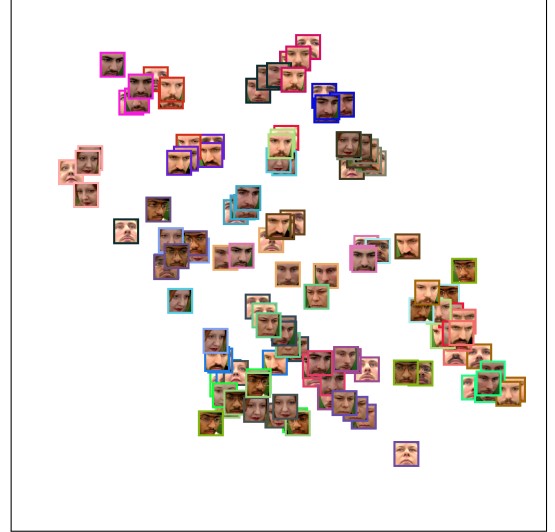 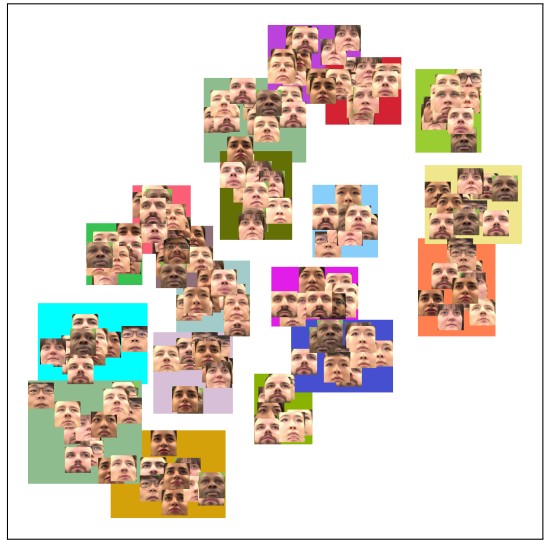

(a) Representations after applying rotation matrices, i.e., $\bar{z}$

(b) Representation obtained from the output of encoder, $z = \mathrm{E}(\cdot)$

Figure 5: **t-SNE visualization.** Qualitative visualization of gaze representations in 2-dimensional space using the t-SNE algorithm. (a) shows the visualization of projection embeddings for multi-view images obtained after applying rotation matrices, i.e., $\bar{z}$. The images with the same timestamp for all four views are highlighted using the same border color. (b) depicts representations for the output of encoder network, i.e., $z = \mathrm{E}(\cdot)$ obtained for images from single camera viewpoint. Best viewed in color and after zooming.

and Ji, 2009). Appearance methods perform better than traditional model methods in real-world settings (Hansen and Ji, 2009; Zhang et al., 2015).

Recent progress in appearance methods relies heavily on deep learning to map eye/face images to gaze directions (Zhang et al., 2015, 2017). Furthermore, a few gaze methods are hybrid. For instance, Park et al. (2018) extracted the relevant eye landmarks from the images and then used these features to train gaze estimators. Other than eye images, several works (Krafka et al., 2016; Cheng et al., 2020b; Fischer et al., 2018a; Cheng et al., 2020a) exploit both eyes and face images in computing gaze direction. Nevertheless, both appearance-based methods and hybrid extensions require a huge amount of labeled data to achieve their potential in terms of accuracy.

As a result of huge label dependence by appearance methods, efforts have been made in the direction of few-shot gaze estimation. In particular, Liu et al. (2018) exploit a two-branch network to predict differential gaze between two images and use a few calibration samples during inference. Furthermore, (Park et al., 2019; Zheng et al., 2020) disentangle gaze from other nuisance factors via training an encoder-decoder architecture (Hinton et al., 2011) to learn gaze specific representations. Recent approaches (Wang et al., 2022; Bao et al., 2022; Jindal and Wang, 2021) leverage labeled source domain and unsupervised domain adaptation for improving the performance of gaze estimation task on the target domain. In a similar spirit, our work attempts to reduce the amount of required label information

via learning gaze representations without relying on gaze labels and utilizing multi-view data (Park et al., 2020; Zhang et al., 2020).

**Self-Supervised Learning.** The goal of self-supervised representation learning is to learn good visual representations from a large collection of unlabeled images. Earlier works in SSL (Zhang et al., 2016; Noroozi and Favaro, 2016; Noroozi et al., 2017; Doersch et al., 2015) used pretext tasks to learn generalizable semantic representations. Some of the recent works (Misra and Maaten, 2020; He et al., 2020; Chen et al., 2020a,b; Caron et al., 2020; Chen and He, 2021; Grill et al., 2020) have shown great success on several vision tasks, e.g, image classification (Caron et al., 2018; Dangovski et al., 2022), object detection (Crawford and Pineau, 2019), semantic segmentation (Moriya et al., 2018), and pose estimation (Rhodin et al., 2018). The recent work by Spurr et al. (2021) extends SSL to hand pose estimation through geometric equivariance representations. Tian et al. (2020) propose to use more than two views to learn invariant representations through contrastive learning.

Recently, a few unsupervised methods have been proposed to learn gaze-specific representations. Specifically, Yu and Odobez (2020) exploit the gaze redirection task to train a gaze estimation model using paired eye images of the same subject. Similarly, Sun et al. (2021) proposed a cross-encoder method to utilize patches of left and right eye images of the same subject as the self-supervised signal. Gideon et al. (2022) is an extended version of Sun et al. (2021) utilizing multi-view images and learning features representing head pose and relative gaze to improve in-domain few-shot gaze estimation performance. However, unlike our work, these methods employ an encoder-decoder framework and thus require a relatively large number of parameters. Also, contrastive SSL approaches are computationally efficient compared to generative SSL approaches (Liu et al., 2021).

## 5. Conclusion

We presented *GazeCLR*, a contrastive learning framework for gaze representations using multi-view camera images. Our framework induces invariance and equivariance properties simultaneously in the learned representations and is effective for gaze estimation task in various settings. Furthermore, we showed that *GazeCLR* representations have the potential to be effective across different domain datasets using only a few calibration samples. *GazeCLR* is a general framework for equivariant representation learning and thus can be explored in the future for other geometry-based applications such as head pose estimation.

## Acknowledgments

This research was supported by the National Eye Institute of the National Institutes of Health (NIH) under award number R01EY030952-01A1. The content is solely the responsibility of the authors and does not necessarily represent the official views of the National Institutes of Health. We thank Mohit Yadav for the helpful discussions.

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

## Appendix A. Definition of Equivariance of Gaze Direction w.r.t. Viewpoints

This section elaborates the equivariance relationship between gaze directions in multi-view geometry, which is also the key idea for *GazeCLR* training framework. Given a specific timestamp in a video, let two samples from different camera viewpoints with gaze directions be $g_{v_1}$ and $g_{v_2}$ in their original respective camera reference system, then the relation between these two gaze directions through their relative camera pose (i.e., $R_{C_1}^{C_2}$), can be given as follows:

$$
\begin{aligned}
g_{v_2} &= R_{C_1}^{C_2} g_{v_1} \\
g_{v_2} &= R_S^{C_2} R_{C_1}^{S} g_{v_1} \\
(R_S^{C_2})^{-1} g_{v_2} &= (R_S^{C_2})^{-1} R_S^{C_2} R_{C_1}^{S} g_{v_1} \\
R_{C_2}^{S} g_{v_2} &= R_{C_1}^{S} g_{v_1} \\
\bar{g}_{v_1} &= \bar{g}_{v_2}
\end{aligned}
\tag{5}
$$

where $R_S^{C_i}$ is relative pose between camera view $i$ and screen. Hence, we follow similar relationship, i.e., $R_{C_2}^{S} g_{v_2} = R_{C_1}^{S} g_{v_1}$, for embeddings obtained from multi-view learning branch and minimizing $L^E$ (Equation 2) will yield $\bar{z}_{v_1} = \bar{z}_{v_2}$. This relation is shown as rotation symbol in Figure 2.

## Appendix B. Additional Results

### B.1. Further Transfer Learning Evaluation

To further evaluate the transferable capability of learned representations obtained from *GazeCLR* framework, we use *Finetuning (FT)* protocol. Here, we fine-tune the entire network (including the encoder) in an end-to-end manner on the target dataset using a few calibration samples from the test subject, and evaluate on the remaining samples.

Table 3: **Transfer Learning Evaluation (Finetuning).** Comparison of various baselines for the *Finetuning* experimental protocol on multiple few-shot settings, for both MPIIGaze and Columbia. Here, we fine-tune whole end-to-end network and utilize few calibration samples during test time. The errors are computed from 10 runs and reported as ($mean^{\pm std}$).

| | MPIIGaze | | | | | | |
|---|---|---|---|---|---|---|---|
| **Method** | 1 | 3 | 5 | 9 | 15 | 50 | 64 |
| w/o Pre-training (Chen and Shi, 2020) | $5.57^{\pm 1.60}$ | $4.65^{\pm 0.71}$ | $4.40^{\pm 0.40}$ | $4.22^{\pm 0.27}$ | $4.13^{\pm 0.17}$ | $4.00^{\pm 0.04}$ | $4.00^{\pm 0.04}$ |
| Autoencoder | $5.65^{\pm 1.60}$ | $4.69^{\pm 0.76}$ | $4.42^{\pm 0.45}$ | $4.16^{\pm 0.21}$ | $4.10^{\pm 0.16}$ | $3.97^{\pm 0.05}$ | $3.96^{\pm 0.04}$ |
| Novel View Synthesis (Rhodin et al., 2018) | $5.53^{\pm 1.32}$ | $4.75^{\pm 0.63}$ | $4.46^{\pm 0.40}$ | $4.27^{\pm 0.25}$ | $4.17^{\pm 0.15}$ | $4.06^{\pm 0.04}$ | $4.06^{\pm 0.04}$ |
| BYOL (Grill et al., 2020) | $5.71^{\pm 1.63}$ | $4.71^{\pm 0.66}$ | $4.35^{\pm 0.31}$ | $4.22^{\pm 0.21}$ | $4.11^{\pm 0.15}$ | $4.01^{\pm 0.05}$ | $4.00^{\pm 0.04}$ |
| SIMCLR (Chen et al., 2020a) | $4.87^{\pm 1.51}$ | $3.93^{\pm 0.54}$ | $3.74^{\pm 0.35}$ | $3.57^{\pm 0.24}$ | $3.47^{\pm 0.12}$ | $3.39^{\pm 0.04}$ | $3.38^{\pm 0.03}$ |
| **GazeCLR (Equiv)** | $\mathbf{4.70}^{\pm 1.49}$ | $\mathbf{3.77}^{\pm 0.51}$ | $\mathbf{3.51}^{\pm 0.32}$ | $\mathbf{3.39}^{\pm 0.18}$ | $\mathbf{3.33}^{\pm 0.11}$ | $\mathbf{3.25}^{\pm 0.03}$ | $\mathbf{3.24}^{\pm 0.02}$ |
| **GazeCLR (Inv+Equiv)** | $4.72^{\pm 1.33}$ | $3.93^{\pm 0.54}$ | $3.68^{\pm 0.34}$ | $3.54^{\pm 0.19}$ | $3.44^{\pm 0.11}$ | $3.37^{\pm 0.03}$ | $3.35^{\pm 0.03}$ |
| | Columbia | | | | | | |
| w/o Pre-training (Chen and Shi, 2020) | $6.96^{\pm 0.55}$ | $5.73^{\pm 0.20}$ | $5.38^{\pm 0.14}$ | $5.23^{\pm 0.09}$ | $5.13^{\pm 0.05}$ | $5.04^{\pm 0.08}$ | $5.00^{\pm 0.09}$ |
| Autoencoder | $7.00^{\pm 0.57}$ | $5.79^{\pm 0.18}$ | $5.49^{\pm 0.15}$ | $5.24^{\pm 0.07}$ | $5.15^{\pm 0.04}$ | $5.03^{\pm 0.08}$ | $5.03^{\pm 0.07}$ |
| Novel View Synthesis (Rhodin et al., 2018) | $7.38^{\pm 0.60}$ | $6.05^{\pm 0.22}$ | $5.78^{\pm 0.14}$ | $5.51^{\pm 0.05}$ | $5.43^{\pm 0.06}$ | $5.33^{\pm 0.06}$ | $5.27^{\pm 0.08}$ |
| BYOL (Grill et al., 2020) | $6.09^{\pm 0.41}$ | $4.97^{\pm 0.22}$ | $4.70^{\pm 0.13}$ | $4.55^{\pm 0.09}$ | $4.43^{\pm 0.04}$ | $4.35^{\pm 0.05}$ | $4.34^{\pm 0.06}$ |
| SIMCLR (Chen et al., 2020a) | $4.36^{\pm 0.20}$ | $3.67^{\pm 0.13}$ | $3.44^{\pm 0.07}$ | $3.34^{\pm 0.05}$ | $3.27^{\pm 0.04}$ | $3.21^{\pm 0.04}$ | $3.19^{\pm 0.05}$ |
| **GazeCLR (Equiv)** | $\mathbf{4.34}^{\pm 0.25}$ | $\mathbf{3.60}^{\pm 0.12}$ | $\mathbf{3.42}^{\pm 0.09}$ | $\mathbf{3.30}^{\pm 0.04}$ | $\mathbf{3.26}^{\pm 0.02}$ | $\mathbf{3.17}^{\pm 0.04}$ | $\mathbf{3.17}^{\pm 0.02}$ |
| **GazeCLR (Inv+Equiv)** | $4.54^{\pm 0.24}$ | $3.75^{\pm 0.12}$ | $3.59^{\pm 0.08}$ | $3.45^{\pm 0.05}$ | $3.39^{\pm 0.03}$ | $3.31^{\pm 0.04}$ | $3.31^{\pm 0.04}$ |

In Table 3, we present the results for FT on MPIIGaze and Columbia, where we fine-tune the whole end-to-end network. For this experiment, we adopt architecture from Chen and Shi (2020), where a subject-dependent bias term is learned along with an end-to-end network. 4-fold and leave-one-out (15-fold) evaluation protocols are used for Columbia and MPIIGaze, respectively.

Unlike Chen and Shi (2020), our input is a full face image, and the backbone is a pre-trained encoder. We take a few calibration samples for each subject during inference and estimate the subject-dependent bias term. We evaluate performance on the remaining samples and repeat this calibration for 10 runs for each subject. Table 3 provides mean and standard deviation of angular errors over 10 runs. We compare the performance of our method with other baselines for various few-shot settings. Results demonstrate that our method consistently outperforms all other pre-training baselines, including Chen and Shi (2020) (w/o Pre-training) for all few-shot settings. This indicates the improved generalization capability of our learned representations, particularly on the MPIIGaze dataset. Also, we observe that our method is either superior or competitive with other baselines on the Columbia dataset.

## Appendix C. Ablation Studies

### C.1. Increasing number of views improves pre-training

In Table 4, we demonstrate the effect of increasing number of views used in pre-training stage of *GazeCLR*. For this ablation study, we conducted experiment for cross-dataset under LLT (similar to Fig. 3) and within-dataset (similar to Table 1) settings, shown in Table 4(a) and Table 4(b) respectively. For 2 views, we considered center and right cameras and for 3 views left camera is included. For LLT setting, the difference in *GazeCLR* performance

for 2/3 views and all 4 views is relatively higher, especially with less number of shots. This shows that for smaller $k$, more views are helpful for *GazeCLR*. Similarly, for within-dataset, *GazeCLR* performance deteriorates with 2/3 views compared to 4 views.

## C.2. More data, better pre-training

In Table 5(a), we study the impact of amount of unlabeled data used for the pre-training stage of *GazeCLR* framework. We observe that the representations learned by *GazeCLR* benefit from more training data and help in generalizing across different domain datasets.

## C.3. Larger batch-size is useful

Next, we vary the batch size to analyze its effect on pre-training, for which results are shown in Table 5(b). We notice that the larger batch size considerably impacts the quality of representations and improves the performance significantly. This observation is consistent to previously observed findings in the self-supervised learning literature (Chen et al., 2020a; He et al., 2020).

Table 4: **Ablation on increasing number of views.** Within-dataset and cross-dataset (LLT) evaluation with increasing number of views used for pre-training stage of *GazeCLR* on both MPIIGaze and Columbia. The ablation study is performed for *GazeCLR(Equiv)* method and evaluation metric is mean angular error (MAE) in degrees, average over 10 runs.

(a) LLT Cross-dataset evaluation

| Dataset | # of views | $k = 20$ | $k = 50$ | $k = 64$ |
|---------|------------|----------|----------|----------|
| MPIIGaze | 2 | 8.94 | 7.59 | 7.25 |
| Columbia | 2 | 7.63 | 4.58 | 4.02 |
| MPIIGaze | 3 | 8.38 | 7.09 | 6.78 |
| Columbia | 3 | 7.20 | 4.45 | 3.88 |
| MPIIGaze | 4 | 8.16 | 7.15 | 6.85 |
| Columbia | 4 | 6.80 | 4.46 | 3.90 |

(b) Within-dataset evaluation

| # of views | MAE (degrees) |
|------------|---------------|
| 2 | 7.72 |
| 3 | 7.06 |
| 4 | 4.83 |

Table 5: **Ablation Study.** 20-shot *linear layer training* for the cross-data gaze estimation on MPIIGaze and Columbia, for two different ablation settings. Ablations are performed for the *GazeCLR(Equiv)* method and evaluation metric is mean angular error (MAE) in degrees.

(a) Varying amount of pre-training data

| Pre-Train Data | MPIIGaze | Columbia |
|----------------|----------|----------|
| MiniEVE | 11.25 | 9.63 |
| EVE | **8.16** | **6.80** |

(b) Varying batch-size used for pre-training

| Batch size | MPIIGaze | Columbia |
|------------|----------|----------|
| 32 | 12.21 | 12.83 |
| 128 | **8.16** | **6.80** |

Table 6: **Ablation Study for mini-batch containing single *vs.* multiple participants.** Within-dataset evaluation under two different types of batches created for the *GazeCLR(Equiv)* method and evaluation metric is mean angular error (MAE) in degrees.

| Task Data | Batch Type | MAE (degrees) |
|-----------|------------|---------------|
| MiniEVE | Single | **4.83** |
| MiniEVE | Multiple | 23.58 |

### C.4. Mini-batch of single *vs.* multiple participants

In Table 6, we experiment with creating batches from single and multiple subjects samples and compare them under within-dataset evaluation (similar to Table 1). We observe that the performance on the gaze estimation task with multiple subject samples was close to the performance of random weights. We hypothesize that this is because in batches with different subjects, negative pairs are easy to classify, given the subject's identity. Therefore, the network has no incentive to focus on gaze information over subject identity.

## Appendix D. Supervised Fine-tuning for Gaze Estimation

In the main manuscript, we demonstrated that the self-supervised gaze representations learned using *GazeCLR* can perform well on a variety of settings when finetuned on the target dataset. Here, we investigate on how performance varies with respect to the amount of data available for finetuning. We evaluate for the within-dataset gaze estimation using *linear layer training* protocol, starting from 10% of EVE training dataset, and gradually increasing to 100%. We compare *GazeCLR(Equiv)* and *GazeCLR(Inv+Equiv)* against "w/o Pre-training" baseline with random initialization, as shown in the Figure 6. *GazeCLR* outperforms the baseline in all training set sizes. It is worth noting that the *GazeCLR* approach only requires 20% of training data to match the performance of the "w/o Pre-training" baseline with 100%. Furthermore, notice that the gap between the performance of *GazeCLR* and baseline decreases as training dataset size increases, showing that *GazeCLR* is effective for training with a few samples.

## Appendix E. Implementation Details for Baseline Methods

We provide further details of our implementation for the pre-training baselines, namely, Autoencoder and Novel View Synthesis (Rhodin et al., 2018).

**Autoencoder.** We use same encoder layers as the *GazeCLR* framework for a fair comparison. The decoder is implemented using DenseNet (Huang et al., 2017a) architecture by replacing convolutional layers with deconvolutional layers of stride 1. The average pooling layer of transition layers is replaced by $3 \times 3$ deconvolutions (with stride 2). The decoder consists of 5 dense blocks, where each block has 4 composite layers with a growth-rate of 32. The compression factor is set to 1.0. All layers are implemented using instance normalization (Ulyanov et al., 2016) and leaky ReLU activation functions (with $\alpha = 0.01$). We use SGD optimizer with momentum 0.9, weight decay $5 \times 10^{-4}$, and initial learning rate is 0.003 (which is decayed using cosine annealing scheduler (Loshchilov and Hutter, 2016)).

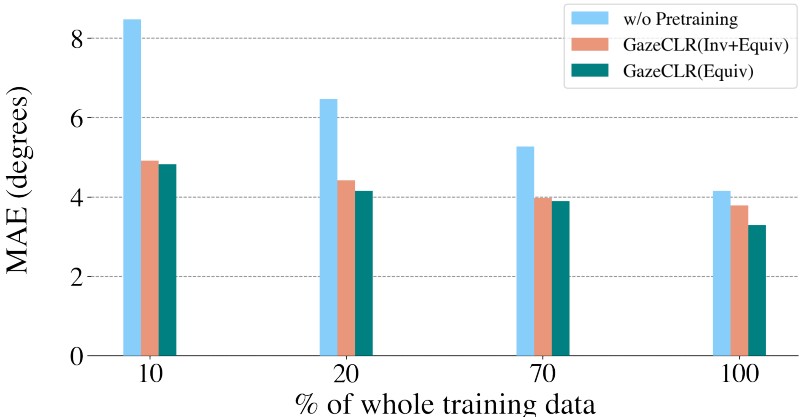

Figure 6: Comparison of the gaze estimation performance for within-dataset using *Linear Layer Training* protocol, versus different % of the labeled training data.

The batch size is 24 and the model is trained for 200K iterations. For inference, we remove decoder layers, and use encoder only for the task of gaze estimation.

**Novel View Synthesis (Rhodin et al., 2018).** This work originally was proposed for 3D human pose estimation task and aimed to learn novel view synthesis, where separate representations for body's 3D geometry ($\mathbf{L}^{3D}$), appearance ($\mathbf{L}^{app}$), and background ($\mathbf{B}$) are trained. For a fair comparison, we train novel view synthesis framework on our dataset using the same encoder architecture as in the *GazeCLR* framework. The decoder layers are same as that of autoencoder baseline. The dimension of appearance-based code ($\mathbf{L}^{app}$) is 32 and of 3D geometry code ($\mathbf{L}^{3D}$) is 480. We ignore the background factor ($\mathbf{B}$) in our implementation, as the EVE dataset has same background across all images. The whole framework is trained using SGD optimizer with learning rate = 0.03, momentum = 0.9, weight decay = $5 \times 10^{-4}$, and cosine annealing for learning rate decay. The training is done for 200K iterations, with the batch size of 16. At each iteration, we randomly sample two views from the EVE dataset, and generate one view image from other view image similar to Rhodin et al. (2018). The trained encoder is then adapted for the gaze estimation, similar to other baselines.

## Appendix F. Additional Visualization

We further qualitatively analyze the relation between learned gaze representations and the ground-truth 2D Point-of-Gaze (PoG). For this, we project gaze representations to 2-dimensional space using t-SNE (Van der Maaten and Hinton, 2008) algorithm and normalize them between 0 and 1. Next, we plot euclidean distance between 2D t-SNE projections and the normalized 2D PoG (dividing by width and height of screen), as shown in Figure 7. The black line in Figure 7 is for the $y = x$ equation. We observe that data is scattered symmetrically around $y = x$, exhibiting a strong correlation (correlation coefficient = 0.623) between gaze representations and ground-truth PoG.

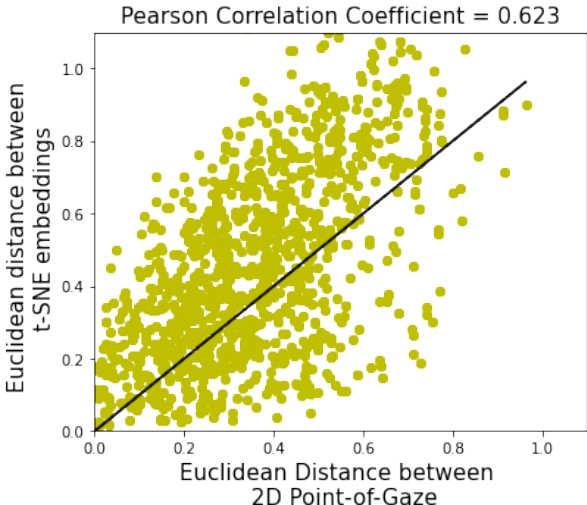

Figure 7: Scatter plot between euclidean distance of normalized 2D PoG and 2D t-SNE projections of gaze representations. The black line is for $y = x$.

