# OpenReview forum: "Contrastive Representation Learning for Gaze Estimation"
_NeurIPS.cc/2022/Workshop/GMML — Gaze Meets ML 2022 Oral_

### Official Review · Reviewer_gBwF · 2022-10-14
**This paper proposes an interesting  self supervised representation learning method that utilizes multiview images for gaze estimation.**

**Rating:** 8
**Confidence:** 4

**Review:**

This paper proposes a self supervised learning method for vision-based gaze estimation. The key novelty is that the proposed method utilizes images captured for the same subject under the same gaze from various cameras for learning equivariance representation. No annotation is required. This method is coupled with invariance representation learning to provide comprehensive representation for gaze estimation. The proposed method was  evaluated on two additional public datasets and showed superior performance using the proposed representation learning method over the  state of the art methods.

The paper is well written with thorough and convincing experimental support.

The results showed that the proposed method with only equivariance representation learning produced comparable performance to the proposed method with both equivariance and invariance representation learning. Is it because equivariance representation also included augmentations used in invariance representation learning?

---

### Official Review · Reviewer_yUHm · 2022-10-17
**Cool addition to the contrastive learning literature**

**Rating:** 9
**Confidence:** 4

**Review:**

A contrastive learning based pretraining strategy is proposed for unsupervised representation learning. The strategy consists of two loss functions - (a) a loss promoting invariance to appearance augmentations, as is done in standard contrastive learning, and (b) a novel loss promoting equivariance to pose, formulated using a dataset containing simultaneous acquisitions of the same person from multiple cameras. I think this is a cool addition to the contrastive learning literature, and shows the power of the contrastive learning framework in molding representation space geometry to mimic dataset-specific characteristics. Experiments are well done, and convincingly showcase the effectiveness of the proposed method.

---

### Meta-Review · Area_Chair_EqyS · 2022-10-20

**Recommendation:** Accept (Oral)
**Confidence:** 5

**Metareview:**

This paper introduces a contrastive self-supervised learning method for gaze estimation, with a novel pose-equivariant loss component. Reviewers have positively commented on the extensive and convincing experiments that demonstrate the effectiveness of the proposed method, in addition to the audience's interest in self-supervised methods. There is one question raised by a reviewer, which I hope the authors can consider addressing on their camera-ready version. Overall recommending accept.

---

### Decision · Program_Chairs · 2022-10-20

Accept (Oral)